# Relationship between Urban Environmental Components and Dengue Prevalence in Dhaka City—An Approach of Spatial Analysis of Satellite Remote Sensing, Hydro-Climatic, and Census Dengue Data

**DOI:** 10.3390/ijerph20053858

**Published:** 2023-02-21

**Authors:** A. S. M. Maksud Kamal, Md. Nahid Al-Montakim, Md. Asif Hasan, Mst. Maxim Parvin Mitu, Md. Yousuf Gazi, Md. Mahin Uddin, Md. Bodruddoza Mia

**Affiliations:** 1Department of Disaster Science and Climate Resilience, University of Dhaka, Dhaka 1000, Bangladesh; 2Geoinformatics Laboratory, Department of Geology, University of Dhaka, Dhaka 1000, Bangladesh; 3Directorate General of Health Services, Mohakhali, Dhaka 1212, Bangladesh

**Keywords:** dengue prevalence, urban environmental components, satellite image, census data, Dhaka city

## Abstract

Dengue fever is a tropical viral disease mostly spread by the *Aedes aegypti* mosquito across the globe. Each year, millions of people have dengue fever, and many die as a result. Since 2002, the severity of dengue in Bangladesh has increased, and in 2019, it reached its worst level ever. This research used satellite imagery to determine the spatial relationship between urban environmental components (UEC) and dengue incidence in Dhaka in 2019. Land surface temperature (LST), urban heat-island (UHI), land-use–land-cover (LULC), population census, and dengue patient data were evaluated. On the other hand, the temporal association between dengue and 2019 UEC data for Dhaka city, such as precipitation, relative humidity, and temperature, were explored. The calculation indicates that the LST in the research region varies between 21.59 and 33.33 degrees Celsius. Multiple UHIs are present within the city, with LST values ranging from 27 to 32 degrees Celsius. In 2019, these UHIs had a higher incidence of dengue. NDVI values between 0.18 and 1 indicate the presence of vegetation and plants, and the NDWI identifies waterbodies with values between 0 and 1. About 2.51%, 2.66%, 12.81%, and 82% of the city is comprised of water, bare ground, vegetation, and settlement, respectively. The kernel density estimate of dengue data reveals that the majority of dengue cases were concentrated in the city’s north edge, south, north-west, and center. The dengue risk map was created by combining all of these spatial outputs (LST, UHI, LULC, population density, and dengue data) and revealed that UHIs of Dhaka are places with high ground temperature and lesser vegetation, waterbodies, and dense urban characteristics, with the highest incidence of dengue. The average yearly temperature in 2019 was 25.26 degrees Celsius. May was the warmest month, with an average monthly temperature of 28.83 degrees Celsius. The monsoon and post-monsoon seasons (middle of March to middle of September) of 2019 sustained higher ambient temperatures (>26 °C), greater relative humidity (>80%), and at least 150 mm of precipitation. The study reveals that dengue transmits faster under climatological circumstances characterized by higher temperatures, relative humidity, and precipitation.

## 1. Introduction

Dengue fever is wreaking havoc in Bangladesh, infecting tens of thousands of people every year, mainly in the capital city of Dhaka. The fever was first reported in 1964, and it reached epidemic proportions in 2000 and 2002, resulting in tens of thousands of cases and a hundred deaths (or 1.7% of all cases) [1]. A devastating dengue fever epidemic claimed the lives of a large number of individuals in 2019 alone. According to the World Health Organization, Bangladesh was one of the world’s highest-risk nations for dengue virus infections in 2019. The *A. aegypti* mosquito transmits a virus that causes a severe flu-like sickness known as dengue. More than half of the world’s population is now in danger, which may harm all residents [2].

About 163.05 million people call Bangladesh home, making it one of the world’s most populous nations, with a density of 1115.62 persons per square kilometer [3]. About 14.4 million people call Dhaka home, making it Bangladesh’s most populous metropolis, with an average of 19,444 inhabitants per square kilometer [3]. The metropolis of Dhaka is expanding at a breakneck pace thanks to an unplanned development strategy characterized by low-lying areas and flat terrain. Due to its socioeconomic traits, high population density, and considerable migrant populations from neighboring Bangladeshi cities, the city has a high sensitivity to dengue disease. Dengue fever is more common in Bangladesh during the pre-and post-monsoon season [1]. Pre-monsoon and post-monsoon dengue cases increased by more than 50 percent between 2000 and 2014 but decreased by less than 10 percent between the years 2015 and 2016 [4].

Dhaka, which is known as tropical, hot, and humid, has an annual average temperature of about 28 degrees Celsius (82.4 degrees Fahrenheit). Increased blood-feeding and viral development rates occur more often as temperatures rise, shortening the extrinsic incubation period and aiding the spread of the dengue virus over time and distance [5]. A city’s land-cover regions that absorb sunlight, radiate heat, and retain less water than natural or background or non-urban land cover, resulting in less humidity for evaporation and cooling, are known as urban heat islands (UHI) [6]. As a result of UHI, the region’s land surface temperature was higher than the surrounding area, and there were fewer vegetated areas and water bodies. The temperature rises produced by the UHI phenomenon led to an increase in air pollution, which led to an increase in respiratory, cardiac, and coronary disease deaths. The *A. aegypti* mosquito, which transmits dengue illness, thrives at a temperature range of 28 to 32 degrees Celsius ([6]. According to recent research, *A. aegypti* larval homes are more common in hot, vulnerable areas than in cooler, less vulnerable ones [4,7,8,9,10]. Different parts of the Dhaka city have been identified as urban heat islands (UHIs) by new research based on satellite images [11]. Dengue incidence in Dhaka city has not yet been studied using day or nighttime satellite photos to identify UHI zones in great detail.

Dengue habitat issues need an understanding of current land-use and land-cover (LULC) patterns and urban heat island expansion. Remote sensing and high-resolution satellite earth observatory data, along with revolutionary image processing methods, have effectively monitored and examined the dynamic changes in LULC, LST, and urban heat-island studies. Bangladesh’s central region has a comparatively small number of LULC mapping projects compared to other ecologically susceptible areas. Land surface temperature (LST) is an essential environmental variable that may be readily obtained from thermal remote sensing data. As a result, LST has been extensively employed in the current literature to determine the effect of LULC change in various ecologically susceptible places throughout the globe. The literature suggests that LST is closely linked to LULC indexes such as normalized difference vegetation index (NDVI), normalized difference water index (NDWI), and normalized difference build-up index (NDBI) and that changes in these indices have a major impact on LST [12,13,14]. In metropolitan environments, LST tends to be very variable. In metropolitan locations, it rises when the percentage of impermeable surface and plant mix is increased [15,16]. It is also recognized that socio-economic factors such as traffic and population density are connected with spatial-temporal changes in LST trends [17]. Other studies suggest that there is a positive temporal relationship and substantial synergy between dengue occurrence and climatic factors that can shape dengue outbreaks [18,19,20].

A number of studies have been conducted concerning dengue incidence in different parts of the world. Marthesqarn et al. [21] studied dengue incidence in Singapore and Honduras using different sampling methods and proposed an estimation model. Liu et al. [22] considered a number of perspectives and analyzed dengue spread. Some researchers assessed the risk of dengue in developing countries [23,24,25,26]. Effects of different environmental factors on dengue outbreaks were also assessed [27,28,29]. Dengue incidence in Bangladesh has garnered substantial attention from a good number of researchers. Dourjoy et al. [30] conducted a comparative study based on machine learning. Hossain et al. [31] incorporated seasonal climatic data and formulated a model to forecast dengue cases. Dey et al. [32] showed the potentiality of machine learning algorithms, considering metrological and socio-economic data, to predict dengue epidemics. A number of studies hypothesized the spreading of dengue disease by the mobility of dengue-infected people from house to house, localities, and by the association of public transport network with the urban socioeconomic factors [33,34,35,36,37,38]. Telle et al. [34] conducted a retrospective study in Delhi City’s dengue incidence and, with both social and environmental risk factors, found higher incidence of dengue in the regions lacking tap water, with higher densities of population, in urban heat islands, and in wealthier regions due to the daily human mobility from higher-dengue-incidence areas to poorer areas. Nakhapakorn et al. [39] obtained a close association of dengue incidence with urban heat islands that was enhanced with the vertical height of buildings in Thailand City. Furthermore, another study showed that the incidence of dengue is related to urban heat islands and exacerbated by global temperature change in tropical to subtropical regions [40].

This research aims to determine whether or not there is a correlation between the incidence of dengue outbreak and the various urban environmental factors. There is various research in the literature relating the environmental components with dengue habitats or prevalence in Bangladesh considering the total number of dengue cases from each city or district levels [30,31,32]. However, no study has focused on the patterns and processes of the urban heat island in relation to dengue habitats in Dhaka City’s local scale. Dhaka City must be examined to fully comprehend the link between urban heat islands and dengue prevalence or habitats. This study used daytime satellite images and reported dengue cases in Dhaka to examine the relationship between urban environmental components (UECs) such as LULC, waterlogging, unplanned structures, UHI, and the occurrence of dengue fever and monitor the prevalence of dengue patients over the last two decades to identify factors that may favor dengue-related mosquito occurrence and virus transmission and thereby facilitate surveillance and prevention. In addition, a dengue risk map was modeled to observe the severity of dengue incidence in the local scale of Dhaka by the various environmental factors and dengue cases, aiming to protect from or reduce future dengue outbreaks.

## 2. Study Area, Materials and Methods

### 2.1. Location

Dhaka City, the capital of Bangladesh, is one of the world’s most populous cities, with a population of 18.89 million [41] (Figure 1). The metropolis is expanding vertically and horizontally to fulfill the needs of its enormous population. As previously described in the literature, this geographical expansion negatively affects LST and microclimatic changes in the metropolitan region. Figure 1 illustrates the location of the research region, namely the Dhaka Metropolitan Area (DMA). The DMA is situated between 23°42′0′′ N and 23°54′0′′ N in latitude and between 90°20′0′′ E and 90°29′0′′ E in longitude. The DMA consists of two city corporations, namely Dhaka South City Corporation (DSCC with 57 wards) and Dhaka North City Corporation (DNCC with 36 wards), and 17 unions in the surrounding region. Packed structures with little open space characterize the DSCC and DNCC. Agricultural land is converted into the bare ground, and the built-up area is expanded to create the periphery. This study does not include restricted regions such as Dhaka Cantonment and the airport area. The urbanization rate of DMA is roughly 4.3%, which has led to the intrusion of urban green areas and wetlands [42]. The climate of Bangladesh is subtropical in the center-north and tropical in the south, with a warm and sunny winter from November through February. Climate-wise, Bangladesh has four different seasons: (1) the dry winter season from December through February, (2) the pre-monsoon hot summer season from March through May, (3) the wet monsoon season from June through September, and (4) the post-monsoon fall season from October through November [43]. The average annual temperature is around 25 degrees Celsius (77 degrees Fahrenheit), and the average annual precipitation is approximately 1300 mm (50 inches). Approximately 44 percent of the DMA comprises higher-temperature zones (27 °C to 30 °C), and these zones are growing by 0.32 °C every decade [44]. The region is bordered by the Tongi Canal (Tongi Khal) in the north, the Turag River in the west, the Balu River in the east, and the Buriganga River in the south. The outer region of the city has a lower population density than the central region.

### 2.2. Materials

Using satellite multispectral remote sensing, hydro-meteorological, and census data of dengue virus-related patients, the relationship between urban environmental components (UEC) such as unplanned structures, waterlogging types, urban heat islands, and dengue mosquito habitats in Dhaka was determined.

Around ten thousand dengue cases have been reported for the whole Dhaka City in the year 2019, of which, the data of about 3154 dengue patients were obtained from eight major hospitals for the purpose of this analysis. The hospitals are Dhaka Medical College and Hospital, Bangabandhu Sheikh Mujib Medical University, Birdem General Hospital, Sir Salimullah Medical College Mitford Hospital, Shaheed Suhrawardy Medical College and Hospital, Combined Military Hospital-Dhaka, Mugda Medical College and Hospital, and Ishtiyaq Medical Center. Patients’ residences in the city of Dhaka were used to geocode dengue cases and create a geodatabase for a GIS environment. Meteorological data for the study area were purchased from the Bangladesh Meteorological Department. The data pack includes daily rainfall, humidity, and atmospheric temperature (maximum, minimum, and average) at three-hour intervals. Using Bangladesh Bureau of Statistics (BBS) census data, a GIS environment generated a population density map of Dhaka City. Multispectral medium-resolution Landsat satellite pictures were used to generate land-use and land-cover maps, showing Dhaka’s flooded regions. Landsat 8 TIRS thermal infrared data were utilized to recover land surface temperature and examine UHI case locations throughout the day and night in the city of Dhaka. The satellite images were collected from the United States Geological Survey (USGS) archive (Table 1). For the comparison and verification of the image-processed results, ground truth data were collected through an extensive fieldwork.

### 2.3. Methodology

For the purpose of this research, a combination of GIS and remote sensing techniques was used. The core procedures analyzed the satellite images to calculate the LULC, LST, NDWI, and NDVI. Later, using the geocoded dengue cases and population density, the kernel density layer was created, which was used to generate a dengue risk map of Dhaka (Figure 2).

#### 2.3.1. Image Preprocessing

Radiometric and atmospheric corrected images were obtained from the USGS archive. Further, we applied the DOS1 mode for visible and near-infrared bands (OLI/2B) and the automated thermal atmospheric correction tool of ENVI package for TIRS bands in this study. Following the completion of the subset of the research area with the Dhaka City shapefile and the masking procedure, the area was partitioned in preparation for further processing and interpretation.

#### 2.3.2. NDVI Calculation

NDVI is one of the most commonly used numerical indicators that employs the electromagnetic spectrum’s visible (VIS) and near-infrared bands. In order to classify land cover, one may compute the normalized difference vegetation index [45]. The NDVI value is always between −1 and +1. According to the Landsat 8 data user manual (2015), the NDVI is computed as follows:(1)NDVI=(NIR−RED) / (NIR+RED)
where red and NIR stand for the spectral reflectance measurements acquired in the red (visible) and near-infrared regions, respectively.

#### 2.3.3. Retrieve LST

The radiative skin temperature of the earth is known as the land surface temperature (LST). The following equations were computed in order to determine the land surface temperature (LST):

**Radiance conversion from DN values:** The distribution of pixel values in an image is called digital numbers, or DN numbers. It ranges from 0 to 255. In most of the imagery, we worked with the DN representatives. Radiance most often has units of watt/(steradian/square meter). Landsat Level-1 data can be converted to TOA spectral radiance using the radiance rescaling factors in the MTL file [46]:(2)Lλ=MLQcal+AL
where

*Lλ* = TOA spectral radiance (Watts/(m^2^ * srad * μm); 

*ML* = Band-specific multiplicative rescaling factor from the metadata (Radiance_Multi_Band_x, where x is the band number); 

*AL* = Band-specific additive rescaling factor from the metadata (Radiance_Add_Band_x, where x is the band number); 

*Qcal* = Quantized and calibrated standard product pixel values (DN).

**Land surface brightness temperature (LST_B_):** The study of urban climate has made extensive use of TIR remote sensing methods. The sensor measures radiance, which is then converted to BT (Brightness Temperature). UHI and surface energy fluxes, for example, are well suited for BT’s broad range of use in landscape characterization [47]. In order to convert spectral radiance to top-of-atmosphere brightness temperature, the MTL file’s thermal constants is used (Table 2).
(3)T=K2 / ln (K1/Lλ+1)
where 

*T* = Top of atmosphere brightness temperature (K).

where

*Lλ* = TOA spectral radiance (Watts/(m^2^ * srad * μm); 

*K1* = Band-specific thermal conversion constant from the metadata (*K1*_Constant_Band_x, where x is the thermal band number); 

*K2* = Band-specific thermal conversion constant from the metadata (*K2*_Constant_Band_x, where x is the thermal band number).

The *K1* and *K2* constants for the Landsat sensors are provided in the following table. It is important to note that we only used band 10 from Landsat 8 due to the larger calibration uncertainty associated with TIRS band 11.

**Retrieve vegetation (PV) coverage from NDVI:** The proportion of vegetation (*P_v_*), which is highly related to the NDVI, and emissivity (ε), is calculated using Equation (4) [46]:(4)$$P_{v}=\left(\left(NDVI\;–\;{NDVI}_{min}\right)\;/\;\left({NDVI}_{max}\;–\;{NDVI}_{min}\right)\right)²$$
where

*P_v_* = Proportion of vegetation;

*NDVI* = DN values from NDVI image;

*NDVI_min_* = Minimum DN values from NDVI image;

*NDVI_max_* = Maximum DN values from NDVI image.

**Retrieve land surface emissivity:** Emissivity can be calculated by the Equation (5) [46]:(5)ε=0.004 ∗ Pv+0.986
where

*ε* = Emissivity of the land surface;

*P_v_* = Proportion of vegetation.

The value of 0.986 corresponds to a correction value of the equation.

The final step is calculating LST using the brightness temperature (BT) of various Landsat satellite thermal bands and LSE derived from *P_v_* and NDVI [48]. LST can be retrieved using Equation (6) [46]:(6)LST=(BT / (1+(λ ∗ BT / ρ) ∗ Ln (ε)))−273.15
where

*LST* = Land surface temperature in Celsius (°C);

*BT* = Sensor brightness temperature in (°C);

*Λ* = Wavelength of thermal band of various Landsat satellite;

*Ԑ* = Emissivity of the land surface;

*ρ* = (h × (c/σ)), which is equal to 1.438 × 10^−2^ mK.

In this case, σ is the Boltzmann constant (1.380649 × 10^−23^ J/K), and h is Plank’s constant (6.62607015 × 10^−34^ J.s and c is the velocity of light (3 × 10^8^ m/s).

## 3. Results

Dhaka’s dengue outbreak of 2019 was sporadic, with cases appearing in different city sections. Uttara, Mirpur, Pallabi, Kafrul, and Kurmitola were the areas with the highest number of cases of dengue fever falling into the category of in the city of Dhaka (Figure 3) (Appendix A). The city’s other areas with a moderate number of patients were Darus Salam, Adabor, and Mohammadpur. A dense concentration of dengue cases in Hazaribagh, Lalbagh, Chakbazar, Bangshal, Kamrangirchar, and Shyampur put these into the high-risk zone. The city’s other areas with a moderate number of patients were Darus Salam, Adabor, and Mohammadpur. Despite this, the remainder of the research region has a reasonably even dengue prevalence, putting them mostly into the very-low- to low-risk class.

### 3.1. Dengue Incidence by LULC

Dhaka is developing to satisfy the requirements of a rising population. The LULC results reveal that in recent decades, uncontrolled development and building have reduced the extent of aquatic bodies. Research showed the city had 8.45% and 5.7% water bodies between 1989 and 2005 [11], which was reduced to 2.51% by 2021. The research region has vegetation cover, representing 12.81% of the total area (Figure 3b). The east of the city has greater greenery. Dhanmondi Lake Park, Mirpur Zoo, Ramna Park, Sohrawardy Uddin, Uttara, etc., are densely vegetated areas. The city’s north, west, south, and center have less plant cover. Due to urbanization and industry, vegetation has diminished drastically in the previous 40 years. In contrast, urban elements are on the rise, reaching up to 82.02% at present. The city is growing vertically and laterally. Buildings, roads, and infrastructures have multiplied throughout the metropolis. Thus, urban areas cover most of the research region. Mirpur, Pallabi, Kafrul, Adabor, Dhanmondi, Mohammadpur, Hazaribagh, Lalbagh, Kamrangirchar, Sutrapur, Shyampur, Motijheel, Rampura, Badda, Gulshan, and Uttara have the most spectral urban traits. In our research, bare lands are uncultivated and sand-filled regions. The city has little bare ground, which represents 2.66% (Table 3) of the total land owing to development. Few barren areas are sand-filled land covers, while others are seasonal owing to the inter-cultivation stage.

The mosquito *A. aegypti* breeds in fresh water. The city’s existing dark and polluted water bodies are less responsible for dengue outbreaks than stagnant water bodies. Because the entire sewerage system of the city has been channeled to major water bodies such as lakes and rivers, industrial and household waste has contaminated these water bodies. Low-lying lands are the only land-cover types that have a direct impact on dengue epidemics. These sites with stagnant monsoon water are helpful for the reproduction and spread of *A. aegypti*. Dengue prevalence is also significantly influenced by urban congestion around these land-cover classes. The tiny water basins are dengue mosquito breeding grounds. Inhabitants in these locations are affected by dengue fever.

### 3.2. Influence of UHI in Dengue Distribution

The minimum and maximum land surface temperature of the study area are 21.59 °C and 33.85 °C, respectively. The city’s northwestern and eastern portions have a low-temperature zone of between 22 and 23 degrees Celsius (Table 4). Significant proportions of the city fall under this category. Khilgaon, Badda, Demra, and Uttarkhan all fall within the zone of low temperatures. A small number of regions in Kadamtali, Turag, and Uttara also have low surface temperatures. As predicted, hardly any dengue cases have been reported from this class. The 23–25 °C temperature range is described as a zone of reasonably low temperatures. As a transitional zone between cold and mild temperatures, this class comprises just a tiny fraction of the study area. It encompasses the region of low surface temperature. This species has been found in the regions of Khilkhet, Badda, Aftabnagar, Azipur, and Ramna. Few dengue cases have been confirmed in this class. The temperature zone between 25 and 26 degrees Celsius has been designated as a mild temperature zone. This class is widely dispersed across the city but is most prevalent in the central region, particularly in Gulshan, Khilgaon, Kafrul, Turag, etc. This class has had few dengue cases documented and therefore may be susceptible to a future epidemic. In this research, the 26–27 °C zone is likewise a prominent LST class. It encompasses practically all of the city’s UHIs. This zone is regarded as having a relatively high LST. This LST class has a considerably higher incidence of dengue than that found lower-temperature zones. This class falls within the zone of increased dengue risk.

Current research suggests that urban heat islands (UHI) in Dhaka are highly susceptible to dengue danger (Figure 3c). In a UHI, classes of land cover such as water bodies, woods, and relative atmospheric humidity are in a red alert zone. This research determined that the 27–32 °C LST zone is a zone with high LST. It generates many UHI in the studied region. Urban heat islands maintain a greater land surface temperature throughout the summer, pre-monsoon, post-monsoon, and dry seasons. The difference between LST and the ambient temperature of a UHI is substantial. UHI of Dhaka City accounts for about 30% of the overall study area. In Dhaka’s Uttara, Mirpur, Pallabi, Kafrul, Mohammadpur, Lalbagh, Hazaribagh, Sutrapur, Shyampur, Tejgaon, Rampura, Badda, and Motijheel neighborhoods, UHIs have been identified. These regions offer ideal conditions for the breeding, larval development, blood feeding, and virus transmission of the *A. aegypti* mosquito.

### 3.3. Dengue Incidence by Population Density

Dhaka, the capital city of Bangladesh, has a population of 18.89 million. It is the sixth most populated city in the world. The Dhaka Metropolitan study area is 306 square kilometers in size. Dhaka North City Corporation (DNCC) and Dhaka South City Corporation (DSCC) were established to manage the metropolitan region (DSCC). These two cities have a total of 54 and 37 wards, respectively. Each year, around one million people relocate to this city in quest of better work opportunities, yet very few leave. As a consequence, the pace of population growth here is accelerating.

The city’s population density is of 29,029 inhabitants per square kilometer is comparable to the national average. In Adabor, Bangshal, Darus Salam, Chak Bazar, Kalabagan, Mirpur, Motijheel, Motijheel, Ramna, Rampura, and Sher-e-Bangla, there are at least 40,000 inhabitants per square kilometer. The population density per square kilometer in Bangshal is 1,485,535, and in Kotoali Thana, it is 123,117.

When there are more *A. aegypti* mosquitoes, they have more vector and blood-feeding opportunities. Data analysis revealed the correlation between population density and the frequency of dengue cases. The bulk of these instances occurs in places with a moderate-to-high population density, such as Bangshal, Sutrapur, and Hazaribagh as well as Lalbagh or Mohammadpur, Chak Bazar, and Mirpur (Figure 3d–f). Dengue viruses may be transmitted from person to person by mosquitoes that can fly approximately a mile and a half. Consequently, a few examples were discovered in locations with a lower population density.

### 3.4. Climatic Factors Influencing Dengue Cases

Bangladesh has a sub-tropical warmer, humid climatic condition. The country has six seasons, and each comprises two months. Nevertheless, it has three major seasonal sub-divisions: pre-monsoon (March–May), monsoon (June–October), and post-monsoon (November–February). These seasons are summer, rainy, and winter, respectively. The variation in climatic factors due to the changes in season, such as ambient temperature, surface temperature, rainfall, humidity, etc., has a significant influence on the dengue outbreak.

**Temperature:** The monthly maximum, minimum, and average atmospheric temperatures exhibit a nearly identical pattern (Figure 4) (Appendix A). In 2019, Dhaka’s yearly average temperature was 25.26 degrees Celsius. In 2019, the average monthly temperature in Dhaka ranged from 18.5 °C to 28.3 °C. January’s average temperature was 18.5 °C, while May’s average temperature was 28.3 °C. The temperature difference between these months is 9.8 °C. Thus, May was the warmest month of that year, while January was the coldest. Typically, the temperature in Dhaka City begins to rise in mid-March and reaches its peak between May and July. In 2019, however, the average air temperature peaked during April.

The scorching summer and monsoon averaged roughly 28 degrees Celsius from April through September. From March to September 2019, the average maximum temperature of the research region remained over 30 °C, but it was lower from November to February. From January through April, the average maximum temperature varied from 24.40 °C to 32.70 °C. The study area’s average minimum atmospheric temperature remained lower during dry seasons and remained higher from April to September. From January through April, the average minimum temperature varied from 12.8 °C to 25.9 °C. On 1 January 2019, the individual lowest atmospheric temperature of the research region was measured as 12.4 degrees Celsius. The highest temperature recorded in 2019 was 37 degrees Celsius on April 25, even if the surface temperature and air temperature of any given day vary throughout the day (Figure 5).

**Humidity:** Between the dry, pre-monsoon, and post-monsoon seasons, there is a noticeable difference in relative humidity (Appendix A). From April through October 2019, the average monthly humidity stayed high (Figure 6). The southern and southwestern summer winds bring additional moisture from the Bay of Bengal. The most humid month is July, while the least humid month is March. July 2019 had the highest average monthly relative humidity at 86%. From May to October of that year, the humidity remained over 80%. In March 2019, the lowest monthly average humidity was recorded at 60%. The majority of dengue outbreaks occur during months with greater relative humidity.

**Monthly average rainfall:** The tropical climate is hot, damp, and humid in Dhaka. From mid-April to mid-October, 70–85% of the yearly precipitation is caused by a humid southwest breeze from the Indian Ocean (Figure 7) (Appendix A). In 2019, Dhaka had a decent quantity of yearly precipitation. In 2019, Dhaka City received a total of 1842 mm of precipitation. Here, the precipitation trend is regrettably erratic. The southwest monsoon is typical in Dhaka, Bangladesh. It takes place from June through September. During this era, the city is bombarded by heavy rains. Due to the unpredictability and suddenness of the precipitation during this season, flooding and flash floods have become commonplace. During pre-monsoon (April to May) and post-monsoon (September to October) periods in Bangladesh, storm surges often occur (from September to December). This storm surge also causes severe rains over the whole nation, including the nation’s capital. Summers receive a substantial amount of precipitation, whereas winters receive relatively little.

The month of January is the driest. In January 2019, there was just 6 mm of precipitation, whereas the majority of precipitation falls in June (Figure 7). June had a total of 368 mm of precipitation. June is the wettest month. However, April, May, July, August, and September had plentiful precipitation. These months have a minimum 150 mm rain gauge meter reading. The total precipitation in May and August was 331 mm and 353 mm, respectively.

Analysis of daily precipitation data acquired from BMD reveals that August had the wettest days, while January had the least. There were 23 rainy days in August compared to just one in January 2019. The difference between the wettest month (June) and the driest month (January) is around 362 mm. In the study year, prolonged and maximum daily rainfall occurred more often. These two erratic precipitation patterns create stagnant water bodies and flash floods in various regions of the research area. Urban areas with uncontrolled sewage systems and a lack of planned growth are more susceptible to flash floods. In low-lying locations with insufficient drainage, stagnant water creates optimal and favorable circumstances for dengue epidemics.

## 4. Discussion

This study aimed to demonstrate a correlation between LST, LULC, climatic indicators, census data, and the incidence of dengue in Dhaka. Based on these parameters, the secondary purpose was to build a map of the city’s dengue risk. We correlated reported dengue cases with geographic location, land surface temperature (LST), urban heat island (UHI), and land-use–land-cover (LULC) of the study area and microclimatic factors such as precipitation, humidity, air temperature, and population census data.

Dengue has become a prevalent monsoon illness in Bangladesh, particularly in Dhaka. Most reported dengue cases occurred between June to December 2019 (Figure 8). The monsoon period (June–October) of 2019 comprised almost 90% of the dengue case of the year. Amongst these months, August stands with the highest percentage of dengue occurrence of the year, with 52,636 detected cases, which represents for 52% of the total. A total of 16,253 and 16,858 cases was reported in the month of July and September, representing 16% and 16.5%, respectively. In contrast, pre-monsoon and post-monsoon periods experienced fewer incidents than the monsoon period. During the monsoons, many regions within Dhaka City experience a number of temporary stagnant water bodies due to waterlogging mostly in the densely populated zones, which influences Aedes mosquitos’ breeding, ultimately leading to more dengue cases. A recent study aimed to assess the risk of dengue transmission in Yogyakarta, Indonesia, using a geographical information system (GIS) and the analytical hierarchy process (AHP) method [49]. The study analyzed data on climatic conditions, land-use patterns, and water-storage practices to understand the relative importance of each factor in determining the risk of dengue transmission. The results revealed that the risk of dengue transmission was highest in urban areas, which were positively associated with the presence of stagnant water bodies (i.e., dirty pond/drain/low-lying waterlogged regions in wet season) in and around densely populated regions.

There is a substantial correlation between the incidence of dengue, rainfall, humidity, and temperature in a given region. Figure 9 shows monthly average dengue data (in thousands), precipitation data (cm), dry-bulb temperature data (°C), and relative humidity data (percent) for 2019. This is only a graphic representation of the correlation between dengue incidence and the urban microclimate. It indicates that relative humidity and precipitation increase in mid-March and stay high until mid-September. Additionally, this time maintains a greater ambient temperature. The number of reported dengue cases in 2019 began to increase in early June, peaked in early August, and began to decline in late August.

The current research provides evidence that the heat island effect has a positive impact on the spatial distribution of *A. aegypti* habitats in urban areas. The majority of dengue cases were recorded in urban heat islands (27–32 °C and 26–27 °C) and 26–27 °C. A similar study [7] published in the journal *Parasites & Vectors* aimed to investigate the influence of urban heat islands and socioeconomic factors on the spatial distribution of *A. aegypti* larval habitats in Salvador, Brazil. The researchers collected temperature data, satellite imagery, and socioeconomic data and used a spatial analysis method to assess the relationship between these factors and the presence of *A. aegypti* larvae. The results showed that urban heat islands were positively associated with the presence of *A. aegypti* larvae, with a higher density of larvae found in areas with higher temperatures. This supports the claim that the presence of urban heat islands (UHI) in Dhaka is contributing to increased dengue cases.

Land-use and land-cover patterns in Dhaka have a substantial correlation with dengue prevalence. Cases of dengue were more widespread in areas with less plant cover as compared to urban features such as homes, roads, etc. Greater NDVI values were correlated with fewer incidences of dengue. Areas with a higher NDWI score also saw a more severe epidemic of dengue. Another study [10] used a combination of remote sensing data and statistical methods to analyze the relationship between land use and dengue outbreaks in São Paulo, Brazil, from 2000 to 2014. This study discovered a similar link between higher dengue incidences observed in areas with a higher proportion of green space (such as parks and forests).

There are a number of limitations in this study. The biological and social dimensions of dengue habitats are not included in this study [50,51]. One of the major limitations of this study is that the dengue data have aggregated over time, and yet, both dengue incidence rates and LST change over time. The dengue cluster signature is created by counting the total number of dengue case during the study period for each locality, as dengue incidence happens more in a certain time of a year. Dengue cases are spatially clustered in the poorest areas with high population density, and the speed with which ongoing transmission occurs is not considered here [52]. Diurnal temperature range (DTR) may affect the *A. aegypti* breeding associated with LST, but we did not consider DTR in this study. Stagnant, clean water bodies such as flowerpots, discarded tires, water storage containers, etc., which are mostly the breeding place of *A. aegypti*, were not possible to delineate using satellite images, and therefore, they are out of scope of this study. The investigation’s scope was also limited by the low-resolution satellite imagery available. By using a high resolution (ideally less than 10 m), the accuracy of the landcover findings may be enhanced. The accuracy of a study such as this could be greatly improved by collecting more data from a wider range of hospitals. Furthermore, only reported cased of dengue patients were considered during the study; however, there is a significant number of unreported cases as well, which are not considered. While the study may not reflect the most up-to-date information, by looking at the analysis presented in this research, researchers can identify patterns and trends in the occurrence of dengue fever. Thus, the information can be useful in predicting future events and informing public health policies. The results can also serve as a baseline for comparison with more recent data. This can help identify changes over time and therefore enable enactment of sustainable prevention measures.

## 5. Conclusions

This research serves the purpose of establishing a relationship between LST, UHI, LULC, urban microclimatic variables, population census data, and the incidence of dengue in Dhaka. This will help as a baseline guidance for further investigation on dengue prevalence in Dhaka City. However, broader investigation may deliver a better result. More field works, high-resolution satellite images, broad-scale datasets with more accuracy, and diurnal and night effects on dengue prevalence can be included in a similar investigation. The authority associated with the combat against dengue and health services in Dhaka City may adopt these outcomes for further policy- and decision making. A similar investigation can be conducted in other cities of Bangladesh to find the relationship between dengue prevalence and LULC, UHI, and climatic factors such as rainfall, humidity, temperature, and population density.

## Figures and Tables

**Figure 1 ijerph-20-03858-f001:**
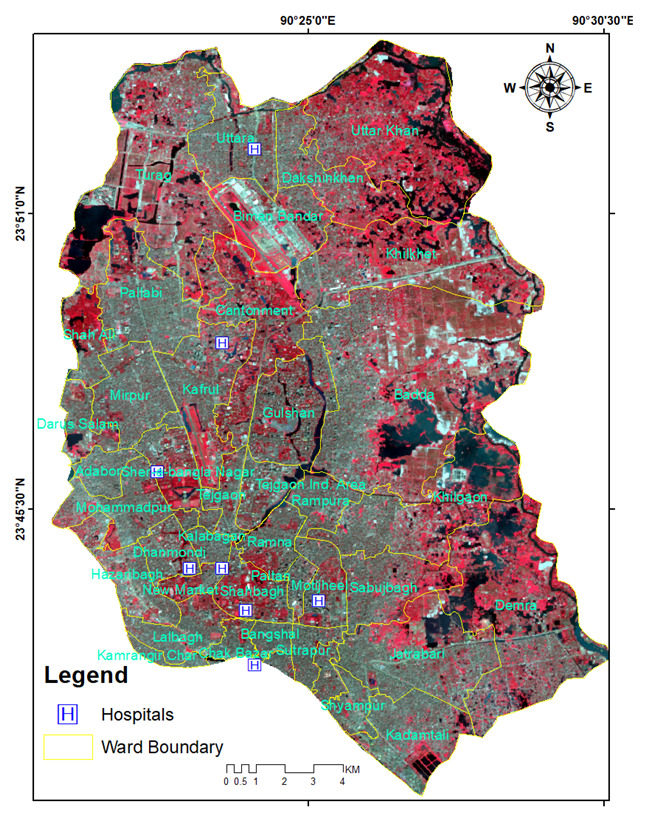
Location of the study area, Dhaka City, Bangladesh (Background image: Landsat 8, 2021: a false color combination map of infrared-band red and green in red, green, and blue colors, where the reddish region indicates vegetated area, gray color urban or built-up area, and dark blue for the water bodies).

**Figure 2 ijerph-20-03858-f002:**
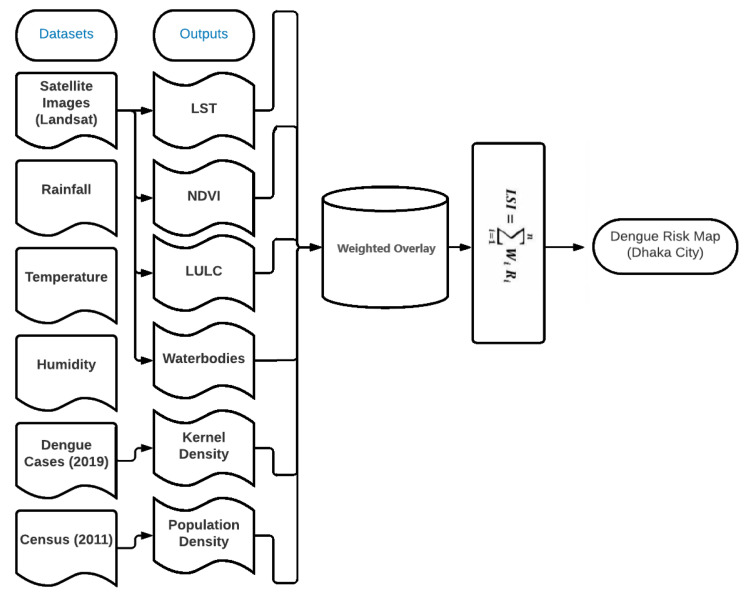
Workflow process involved in this study.

**Figure 3 ijerph-20-03858-f003:**
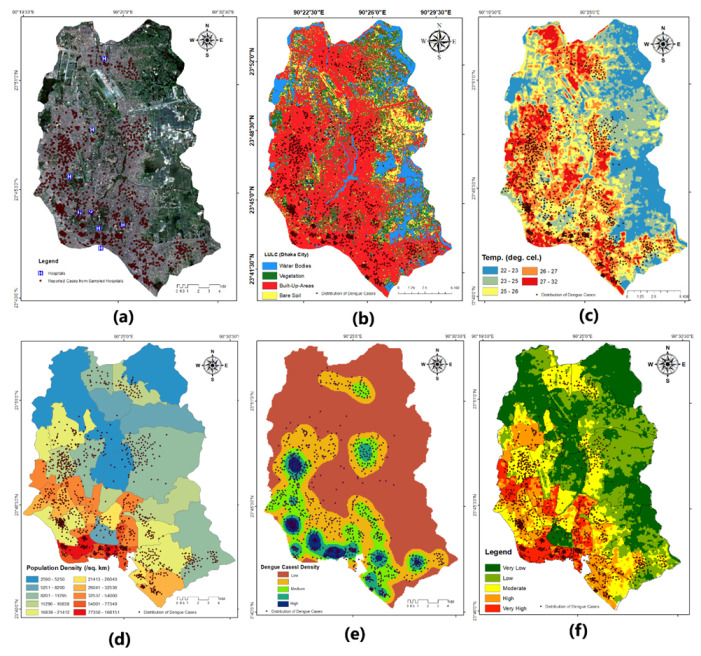
(**a**) Dengue cases location on a true color image of Landsat 8 OLI; red dots show the reported dengue patients in hospitals, (**b**) distribution of Dengue incidence by LULC, and (**c**) influence of UHI in dengue distribution by the ranges of LST; higher-temperature region shows the higher density of reported dengue patients, (**d**) population density in Dhaka City with the number of reported dengue patients distribution, (**e**) dengue incidence density prepared by the interpolation methods, and (**f**) dengue risk map of Dhaka City prepared using the various environmental components; red dots indicate the distribution of reported dengue patients in hospitals.

**Figure 4 ijerph-20-03858-f004:**
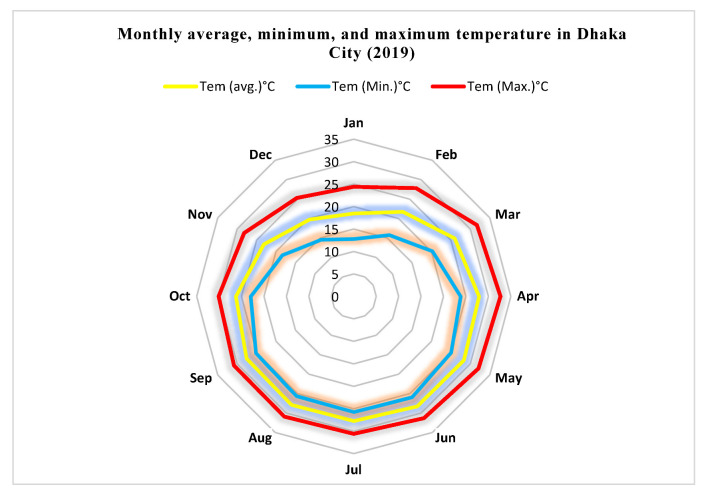
Monthly average, maximum, and minimum temperature in Dhaka City (January to December), 2019.

**Figure 5 ijerph-20-03858-f005:**
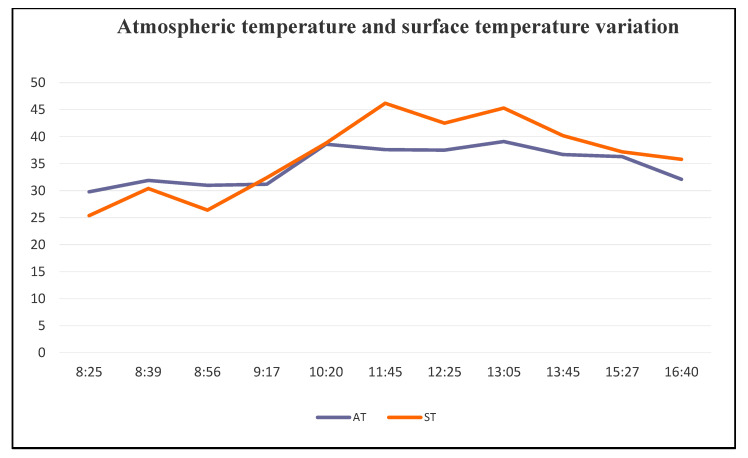
Variation between atmospheric temperature (AT) and land surface temperature (ST) on 29 March 2019.

**Figure 6 ijerph-20-03858-f006:**
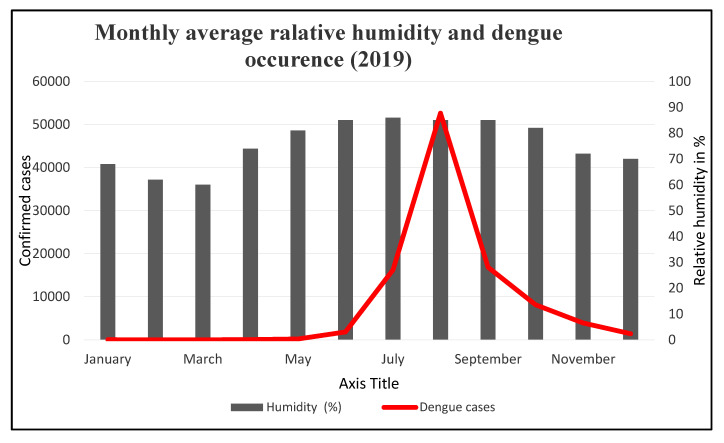
Monthly average relative humidity and dengue cases (1000x) in Dhaka (2019).

**Figure 7 ijerph-20-03858-f007:**
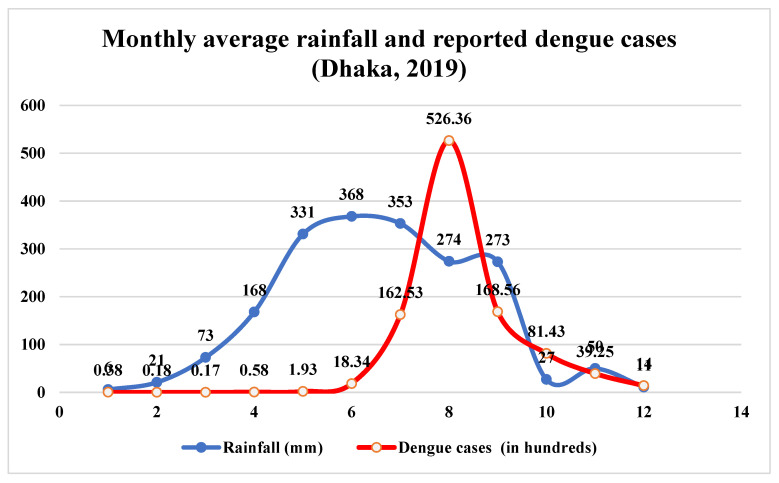
Monthly average rainfall and dengue cases (100×) in Dhaka City (January to December), 2019.

**Figure 8 ijerph-20-03858-f008:**
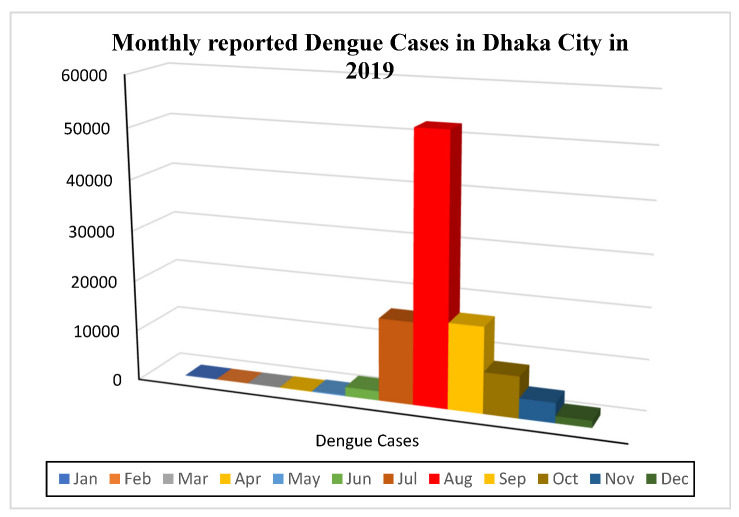
Total number of reported dengue cases in Dhaka City (2001–2019).

**Figure 9 ijerph-20-03858-f009:**
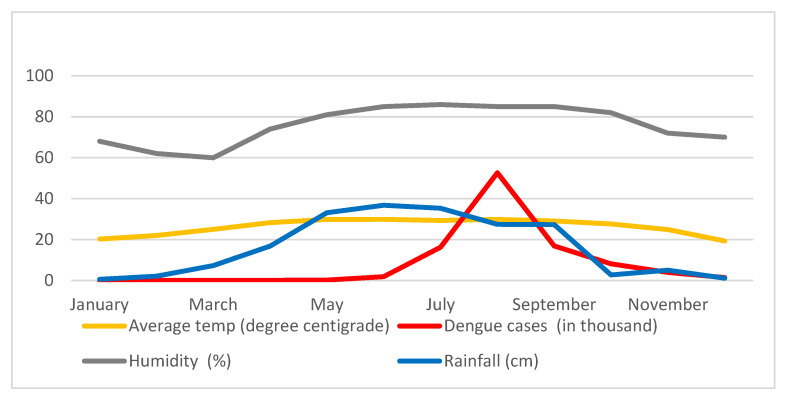
Correlation between dengue cases (1000×), rainfall (cm), average temperature (centigrade), and relative humidity (%).

**Table 1 ijerph-20-03858-t001:** Data sources Satellite Images with their Acquisition Date and Spatial Resolution.

Sensor	Date	Resolution (m)
Landsat OLI/TIRS	9 June 2019	30
Sentinel 2B	17 June 2019	10–20
Sentinel 2B	13 June 2019	10–20

**Table 2 ijerph-20-03858-t002:** Constant K1 and K2 for Landsat Satellites.

Constant	Landsat OLI/TIRS Band-10
K1 (watts)	774.8853
K2 (kelvin)	1321.0789

**Table 3 ijerph-20-03858-t003:** Supervised classes for LULC with their area coverage and total percentage.

Class	Description	Area (sq. Km)	Area (%)
Water Bodies	Rivers, lakes, ponds, etc.	7.99	2.51
Vegetation	Agricultural land, shrubs, dense forests, etc.	40.51	12.81
Bare Soil	Open fields, sand-filled fields, playgrounds, etc.	8.33	2.66
Built-up Areas	Buildings, roads, industries, and development features.	259.17	82.02
Total		316.71	100%

**Table 4 ijerph-20-03858-t004:** LST classes with their range, related LULC classes.

Temperature Class	Range (°C)	Area (%)
Low	22–23	39.6
Moderately low	23–25	22.1
Moderate	25–26	16.6
Moderately high	26–27	13.9
High or UHI	27–32	7.8

## Data Availability

Not applicable.

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
