# Peer review of "Relationship between Urban Environmental Components and Dengue Prevalence in Dhaka City—An Approach of Spatial Analysis of Satellite Remote Sensing, Hydro-Climatic, and Census Dengue Data"

_ijerph, 2023, doi:10.3390/ijerph20053858_

Round 1
Reviewer 1 Report
The authors have shown the impact of several environmental factors related to the Dengue outbreak that occurred in the year 2019 in Bangladesh, specifically the capital of the country was the main focus. However, the following considerations/suggestions/concerns need to be addressed before it is considered for publication.
1. Data availability statement is missing. Please share the Data using any recognized data repository for the reproducibility of this study.
2. The literature review section is not present in the article. Design a dedicated literature review section and present/discuss relevant literature immediately after the introduction.
3. This study only explored the co-relation between several environmental factors and the occurrences of Dengue Incidents in 2019. What is the significance of the current time of this study as it explored data around 3 years ago?
4. In the introduction section, the authors claimed "In addition, a model is created to predict how Dhaka's evolving urban environment may affect dengue outbreaks in the future. " However, I am unbale to find any prediction model in the manuscript.
5. Authors have collected/Purchased different types of data from different stakeholders including BMD, BBS, Dhaka City Hospitals, and so on. All the data should be made available to better understand the pattern and data shape.
6. In the Methodology section, the authors further claimed "This research aims to determine whether or not there is a correlation between the incidence of dengue fever and the various urban thermal environmental factors. "- Only correlation establishment can not be considered as a significant contribution. I suggest highlighting the prime contribution of this research more specifically. A further prediction model development is highly recommended.
7. Following articles can be helpful/cited to better understand about prediction model regarding dengue incidents.
https://journals.plos.org/plosone/article?id=10.1371/journal.pone.0270933
https://journals.plos.org/plosntds/article?id=10.1371/journal.pntd.0008349
https://journals.plos.org/plosone/article?id=10.1371/journal.pone.0270933#
8. Check the article thoroughly for typos like
| Sohrawardy Uddin, Uttarakhand, etc |
Author Response
The authors have shown the impact of several environmental factors related to the Dengue outbreak that occurred in the year 2019 in Bangladesh, specifically the capital of the country was the main focus. However, the following considerations/suggestions/concerns need to be addressed before it is considered for publication.
Authors response: Thank you very much for your vigilant observation and suggestions to improve the manuscript. We agreed with your observation and trying our best to response your observations and corrected accordingly.
- Data availability statement is missing. Please share the Data using any recognized data repository for the reproducibility of this study.
Authors response: Our database is available and added in the supplementary sections of this journal. Due to the data policy of the organization from where we purchased (Hospitals/BMD), it is not possible to deposit in public domain or repository.
- The literature review section is not present in the article. Design a dedicated literature review section and present/discuss relevant literature immediately after the introduction.
Authors response: We agreed with your observation and addressed all the relevant literature review in the introduction section.
- This study only explored the co-relation between several environmental factors and the occurrences of Dengue Incidents in 2019. What is the significance of the current time of this study as it explored data around 3 years ago?
Authors response: Yes, we just trying to understand the relationship between urban environmental components and Dengue incidence in 2019, and prepare a dengue risk map of Dhaka city. By this study, the policy makers or concern authority can take necessary measures to prevent further dengue outbreak in this city. This is the first study using satellite imagery-based outcome to find out the influences of urban environmental parameters for dengue habitats in localities of Dhaka city.
- In the introduction section, the authors claimed "In addition, a model is created to predict how Dhaka's evolving urban environment may affect dengue outbreaks in the future. " However, I am unbale to find any prediction model in the manuscript.
Authors response: We agreed with your observation. To predict the future outbreak is not our current objective rather we tried to find out understand the relationship between urban environmental components and Dengue incidence in 2019. We just wanted to map a dengue risk zone associated with the urban environmental components in 2019. We updated the section you raised.
- Authors have collected/Purchased different types of data from different stakeholders including BMD, BBS, Dhaka City Hospitals, and so on. All the data should be made available to better understand the pattern and data shape.
Authors response: We agreed and supplied the data as supplementary in this journal.
- In the Methodology section, the authors further claimed "This research aims to determine whether or not there is a correlation between the incidence of dengue fever and the various urban thermal environmental factors. "- Only correlation establishment can not be considered as a significant contribution. I suggest highlighting the prime contribution of this research more specifically. A further prediction model development is highly recommended.
Authors response: We agreed and addressed the significance of this study in discussion and introduction section.
- Following articles can be helpful/cited to better understand about prediction model regarding dengue incidents.
https://journals.plos.org/plosone/article?id=10.1371/journal.pone.0270933
https://journals.plos.org/plosntds/article?id=10.1371/journal.pntd.0008349
https://journals.plos.org/plosone/article?id=10.1371/journal.pone.0270933#
- Check the article thoroughly for typos like
|
Sohrawardy Uddin, Uttarakhand, etc |
Authors response: We agreed and updated the references in the literature review. As our current study’s objective is not to predict the future incidence of dengue, we would not like to address this issue here right now. We may work on it in future. We have checked carefully the typo mistake.
Overall, thank you very much for your valuable comments and suggestions to improve the manuscript.

Reviewer 2 Report
I have found the study content is scientifically very interesting and important for country like Bangladesh. I have some observations that I mentioned in the attached file. In brief I would say that the study objective was not mentioned in the introduction section clearly. To observing the correlation between the incidence of dengue fever and the various urban thermal environmental factors and to generate dengue risk map of Dhaka only 1 year (2019) dengue data was considered, scenario can be different in other years. In discussion section it was not mentioned whether this study findings agree or disagree with same type of other previous studies. In discussion section some result was mentioned with figure and bar diagram, it can be placed in result section. Most importantly, a table would be better to understand mentioning the area of Dhaka city that I added to the text to think.
Please see attached the comments in your manuscript file.

Author Response
I have found the study content is scientifically very interesting and important for country like Bangladesh. I have some observations that I mentioned in the attached file. In brief I would say that the study objective was not mentioned in the introduction section clearly. To observing the correlation between the incidence of dengue fever and the various urban thermal environmental factors and to generate dengue risk map of Dhaka only 1 year (2019) dengue data was considered, scenario can be different in other years. In discussion section it was not mentioned whether this study findings agree or disagree with same type of other previous studies. In discussion section some result was mentioned with figure and bar diagram, it can be placed in result section. Most importantly, a table would be better to understand mentioning the area of Dhaka city that I added to the text to think.
Please see attached the comments in your manuscript file.
Authors response: Thank you very much for your vigilant observation and comments. We have addressed carefully all the issues according to your guidelines. The study objective is now clearly mentioned in the manuscript. Due to the epidemic outbreak of dengue cases in the 2019, we just tried to understand the correlation between the incidence of dengue patients and various urban environmental components or factors in the current study, and tried to retrieve a dengue risk map of Dhaka city. There is no previous study in Bangladesh to find out the relationship, but in some other countries like south Africa, Brazil, there are some studies to find out the relationship. We think the spatial distribution of dengue risk map indicates more meaningful information rather giving a table, where the locality boundaries might create more confusion. Anyway, thank you once again for your nice observations and suggestions to improve the manuscript.
We agreed and have addressed all of your raised issues in the pdf version step by step below.
Response to reviewer Comments-
ANA1:
Suggestion for Page 2, Line 65 has been addressed.
ANA2:
Suggestion for Page 2, Line 77 has been addressed.
ANA3:
Suggestion for Page 2, Line 91 has been addressed.
ANA4:
We agree on this point. The analysis was a combination of 2019 reported dengue cases collected from 8 major hospital and satellite images which was not clearly mentioned in the section. This has now been addressed.
ANA5:
We agree on this point. The section has been introduced additional description to make the objective of the research clearer.
ANA6:
The issue with the location map has been addressed.
ANA7:
We agree on this point. The name of the hospitals has been included.
ANA8:
The data chart used in the discussion included the reported cases from all the hospitals in Dhaka city. However, for this study we only considered data from major hospital. But we understand that this can create some confusion for the audiences too, and so we have made necessary correction to the discussion section that now solves the issue.
ANA9:
We agree on this point and have corrected the issue according to your suggestion.
ANA10:
The whole Dhaka city is composed multiple wards and upazilla. Also, the resulted LULC, LST, NDVI, NDWI, Population density, Kernel density, and risk map represents more of a holistic idea with interpolated datasets. Thus, quantifying for each upazilla may create confusion since the results are categorized in multiple classes and more than one class falls under one are in many cases.
ANA11:
In the discussion section we have now compared our result not only with similar studies but also with the secondary information collected.
ANA12:
We agree on this point and have corrected the issue according to your suggestion.
ANA13:
We agree on this point and have corrected the issue according to your suggestion.
In the last page, you suggested to make a table to compare the research outcomes with different factors, but due to complexity to separate local units (e.g., in a same local unit, mix information exist), we think its not feasible or informative enough to understand, rather spatial distribution of the dengue incidence results with various environmental factors might more understandable.
Overall. Thank you very much for your kind and passionate review and suggestions to improve our manuscript.

Reviewer 3 Report
In this paper, various factors related to the incidence of dengue fever in Dhaka City in 2019 were studied. Helping to improve our understanding of dengue transmission in cities.
I recommend ‘minor revision’, some explanations and supplements are needed, please revising follow detail questions.
Question 1: L262-264. Please complete the title of the Figure 3, not just a few phrases. Also, the title of the Figure should describe the elements in more detail, as shown in Figure 4 with the three-color meaning, as well as in Figure 5-7.
Question 2: L285-293. L327-328. Etc. Most of these assumptions about mosquitoes would be more appropriate move to Discussion if they were not analyzed directly.
Author Response
In this paper, various factors related to the incidence of dengue fever in Dhaka City in 2019 were studied. Helping to improve our understanding of dengue transmission in cities.
I recommend ‘minor revision’, some explanations and supplements are needed, please revising follow detail questions.
Authors Response: Thank you very much for your kind review and comments. We encouraged with your valuable comments and have updated the manuscript according to your suggestions.
Question 1: L262-264. Please complete the title of the Figure 3, not just a few phrases. Also, the title of the Figure should describe the elements in more detail, as shown in Figure 4 with the three-color meaning, as well as in Figure 5-7.
Authors Response: Agreed and updated the manuscript.
Question 2: L285-293. L327-328. Etc. Most of these assumptions about mosquitoes would be more appropriate move to Discussion if they were not analyzed directly.
Authors Response: Agreed and updated the manuscript.

Reviewer 4 Report
Dear Authors,
Please go through the text of the manuscript in a critical manner and correct the English grammatical errors and usages. Also, incorporate the expansion of acronyms (NDVI, NDWI, NDBI etc,), when those are used initially for clarity.
If these minor revisions are carried out the manuscript could be improved.
Author Response
Dear Authors,
Please go through the text of the manuscript in a critical manner and correct the English grammatical errors and usages. Also, incorporate the expansion of acronyms (NDVI, NDWI, NDBI etc,), when those are used initially for clarity.
If these minor revisions are carried out the manuscript could be improved.
Authors Response: Thank you very much for your valuable comments and suggestions. Agreed with your observation and checked the manuscript carefully for the English grammatical errors and corrected accordingly. We have also updated the acronyms according to your suggestions.

Round 2
Reviewer 1 Report
The authors have rectified/addressed all the concerns raised.
Author Response
The authors have rectified/addressed all the concerns raised.
Authors Response: Thank you very much for your comments and suggestions to improve the manuscript.
